# Adverse event profile differences among long-acting gonadotropin-releasing hormone analogs: A real-world, pharmacovigilance study

Yuting Chen[1,2°], Weitao Lu[3°], Ruilian Liao[4], Ximin Zhang[5], Wang Chen[1], Jing Wang[1], Huancun Feng[6]*

1 Department of Pharmacy, Guangzhou Red Cross Hospital of Jinan University, Guangzhou, Guangdong, China, 2 School of Pharmaceutical sciences, Southern Medical University, Guangzhou, Guangdong, China, 3 Department of Pharmacy, Jiangmen Central Hospital, Jiangmen, Guangdong, China, 4 Sun Yat-Sen University Zhongshan Ophthalmic Center, Guangzhou, Guangdong, China, 5 Guangzhou Civil Aviation College, Guangzhou, Guangdong, China, 6 Department of Pharmacy, The Third Affiliated Hospital of Southern Medical University, Guangzhou, Guangdong, China

° These authors contributed equally to this work.

* 87733131@qq.com

## Abstract

### Background

Long-acting Gonadotropin-releasing hormone analogs(GnRHa), including leuprolide, goserelin, histrelin, buserelin, triptorelin, have been widely used for a variety of diseases including prostate cancer, breast cancer, endometriosis, uterine leiomyomas, and central precocious puberty (CPP). However, their real-world safety profile differences have not been adequately compared.

### Objective

We aimed to investigate the adverse event (AE) profile differences of long-acting GnRHa reported by the US Food and Drug Administration Adverse Event Reporting System (FAERS).

### Methods

All indications were searched long-acting GnRHa, as primary suspect drugs, from FAERS data (January 2004 to September 2023). We performed disproportionality analyses by reporting odds ratios (ROR) and conducted univariate and multivariate logistical regression analyses to determine the odds ratio (OR) of serious AEs associated with long-acting GnRHa under different exposure factors.

### Results

Reproductive system and breast disorders accounted for the greatest proportion of adverse events among the five long-acting GnRHa formulations analyzed. Both

**Data availability statement:** All relevant data are within the paper and its Supporting information files.

**Funding:** This work was supported by 2024 PSM Guangdong Pharmaceutical Science Research Fund (No.2024KP106) and 2024 Guangdong Provincial universities characteristic innovation project (No.2024KTSCX078). The funders had no role in study design, data collection and analysis, decision to publish, or preparation of the manuscript.

**Competing interests:** The authors have declared that no competing interests exist.

buserelin and histrelin showed distinct adverse effect profiles, with buserelin demonstrating a higher incidence of gastrointestinal disorders and histrelin showing a greater propensity for psychiatric disorders. Logistic regression analysis revealed these five medications carried an elevated risk of significant medical events, and this risk was notably lower in pediatric patients (<18 years) compared to adult populations (≥18 years).

## Conclusions

Significant disparities exist between the adverse event profiles of long-acting GnRHa. The identification of high-risk factors and the enhancement of AEs monitoring are crucial during clinical application.

---

### Introduction

Dysregulation at any level of the hypothalamic-pituitary-gonadal (HPG) axis leads to, or exacerbates, a range of hormone-dependent diseases. Given its crucial role as a regulator of the HPG axis, agonist and antagonist analogues of gonadotropin-releasing hormone (GnRH) have proven efficacy in treating these conditions. Additionally, GnRHa plays a pivotal role in assisted reproductive therapies [1]. GnRHa are available as rapid-acting or long-term depot preparations. The long-acting preparations available include: leuprolide, goserelin, histrelin, buserelin and triptorelin. They were first approved in the following order: buserelin (1984), leuprolide (1985), goserelin (1989), triptorelin (early 1990s), and histrelin (2005). These compounds have a high affinity for the pituitary Luteinizing Hormone-Releasing Hormone (LHRH) receptor and are resistant to enzymatic degradation. Long-acting GnRHa have been commonly used for the medical treatment of prostate cancer, breast cancer, precocious puberty, endometriosis, adenomyosis, also for the prevention of chemotherapy-induced premature ovarian failure in gynecological cancer [2–5].

Although long-acting GnRHa have a wide range of applications, the reported adverse reactions during the treatment of these diseases are also not uncommon (Fig 1). In children with CPP receiving GnRHa therapy, multiple adverse events have been documented in clinical reports, including vaginal spotting/bleeding [6], subcutaneous nodules with sterile abscess formation [7], slipped capital femoral epiphyses (SCFE) [8], pseudotumor cerebri (PTC) [9] and hypertension [10]. When treating prostate cancer, the adverse reactions of long-acting GnRHa reported including life-threatening anaphylaxis [11], follicular mucinosis and mycosis-fungoides-like drug eruption [12], cardiovascular toxicities [13]. Moreover, multiple clinical evaluations have found many adverse reactions containing leukocytoclastic vasculitis [14], vasomotor symptoms and bone mineral density loss [15], low sexual desire, abnormal vaginal bleeding, and liver function damage [16] in the treatment of endometriosis with GnRHa.

In summary, GnRHa exhibit various adverse reactions in the treatment of various indications. However, due to the scarcity of cases, rigorous qualification standards,

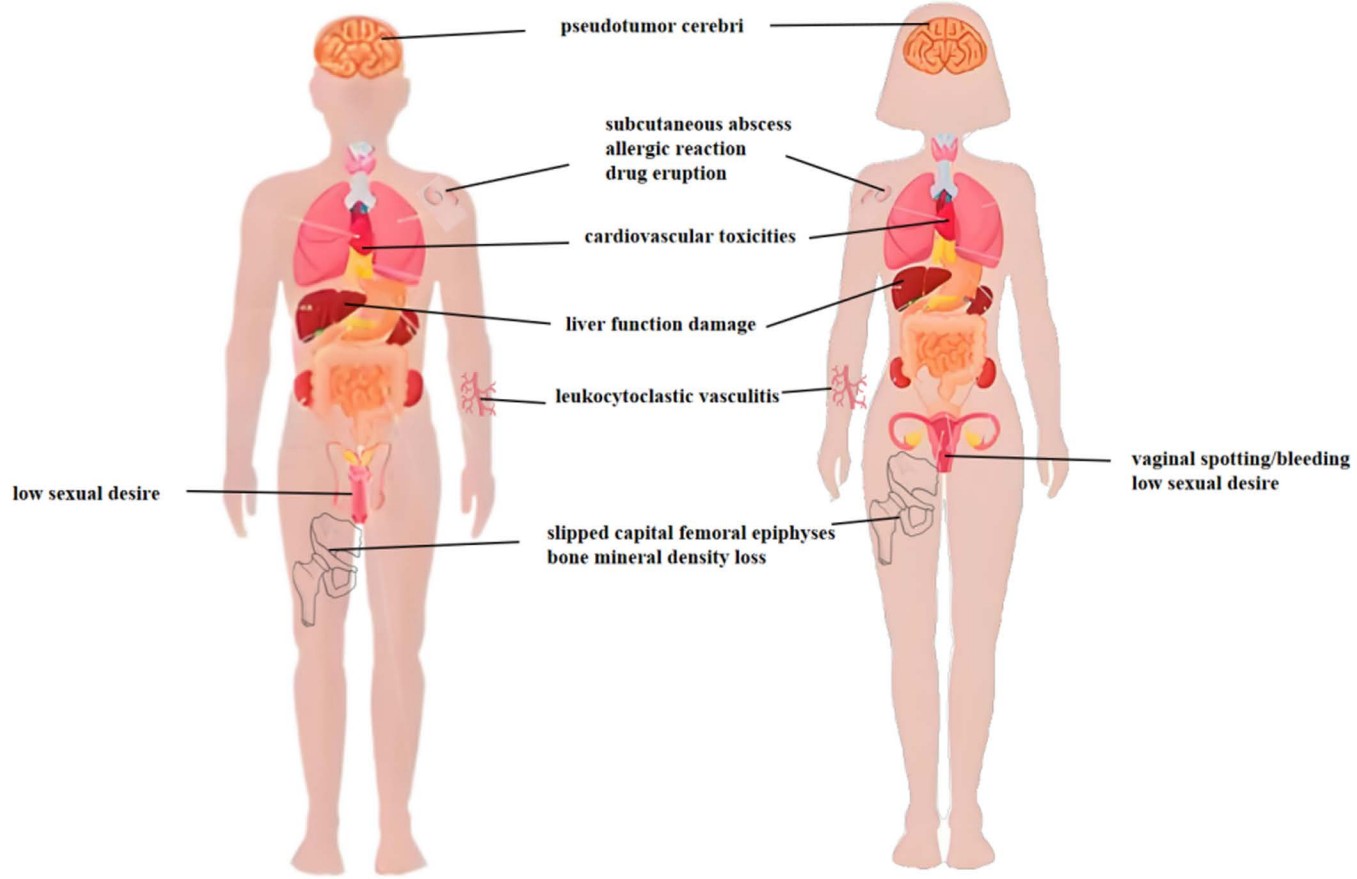

**Fig 1. Adverse reactions reported of Long-Acting GnRH Agonists on Human Organs.** Long-acting GnRHa have been reported corresponding adverse reactions in various organ systems of the human body for different indications. The main affected organs include the pituitary gland, bones, cardiovascular system and reproductive organs, etc.

and constrained duration for monitoring and follow-up, clinical trials or case reports cannot fully reflect the safety profile of long-acting GnRHa in the real-world. FAERS, a public pharmacovigilance database that is utilized for collecting post-marketing safety data submitted to the FDA all over the world, serves as a valuable resource compensating for the limitations of clinical trials. Several studies have utilized this database with comparable methodologies, such as analysis of sildenafil [17], Everolimus [18], topotecan [19], acetylsalicylic acid [20] and a comparison of vinorelbine and vincristine [21]. These precedents support the feasibility and relevance of conducting our research on long-acting GnRHa using the FAERS database. Therefore, we utilized FAERS to evaluate the safety characteristics of long-acting GnRHa and provide a practical and secure reference for their clinical applications.

## Materials and methods

### Data sources

FAERS serves as a post-market safety surveillance program for all medications that have received approval from FDA. We extracted the data utilized in this study from FAERS, which consists of spontaneous reports submitted by consumers, healthcare professionals, and manufacturers worldwide. This retrospective pharmacovigilance study employed a disproportionality analysis based on adverse drug reactions reported in FAERS, one of the prominent public spontaneous

reporting systems encompassing reports from over 100 countries [22]. Serving as a comprehensive post-marketing safety surveillance initiative for FDA-approved drugs and therapeutic biological products globally, FAERS collects spontaneous safety reports and post-marketing clinical studies related to drug use within and outside the United States. The FAERS database utilizes the Medical Dictionary for Regulatory Activities (MedDRA) coding system. In this study, the international standardization of adverse drug events (ADEs) was achieved through the application of system organ class (SOC) and preferred terms (PTs) from MedDRA [23].

### Data mining

In this study, the original data is sourced from the FAERS database, extracted and queried using the OpenVigil 2.1 website (http://openvigil.sourceforge.net/). The time range for data extraction is limited from the establishment of FAERS database to September 2023. As primary or secondary suspected drugs, we conducted a search in the FAERS database, spanning from January 2004 to September 2023, using the keywords "buserelin," "goserelin," "histrelin," "leuprolide," and "triptorelin" to identify all relevant indications. The duplicate case is considered if the case ID, reporting date, and patient characteristics are all the same.

The Reporting Odds Ratio (ROR) analysis was conducted to quantify the statistical linkage between the drug of interest and specific AEs recorded in the FAERS database. The key algorithms leveraged for signal detection are outlined in S1 Table. AE would be deemed highly correlated with the treatment involving the drug of interest if all criteria outlined in S1 Table were concurrently satisfied, with elevated values suggesting a stronger statistical association. Each validated AE was assigned a preferred term (PT) and subsequently categorized into its respective System Organ Class (SOC) utilizing the Medical Dictionary for Regulatory Activities (MedDRA, version 24.1) as a reference.

### Statistical analysis

In this study, the disproportionality method was used for signal mining of ADEs. The disproportionality method is a widely used method in ADR monitoring, which is based on a 2 × 2 league table (four-cell table) to calculate the ratio of observed to expected drug-event values, and an imbalance is defined if it exceeds a prespecified threshold. The reporting odds ratio (ROR) in the disproportionality method is one of the most commonly used frequency methods, which are characterized by simple calculation and good consistency of results [24]. The corresponding algorithms, equations and criteria listed in S1 Table, which were used as the judgment conditions for the signal generation, and the fulfillment of the above conditions suggested that the drug was statistically associated with ADEs. Excel 2019 software was used for data cleaning, mapping between system organ classification (SOC) and preferred term (PT) and statistical analysis.

A descriptive analysis of the demographic characteristics was conducted, and the proportion difference of AE reports among long-acting GnRHa was analyzed using either the chi-square test or Fisher's exact test, both implemented in SPSS 26.0. We executed Disproportionality analyses utilizing ROR. Following the removal of duplicates, we compared the proportion of valid signals within each System Organ Class (SOC) using the chi-square test. The entire process of data processing and analysis is shown in Fig 2.

## Results

### Population characteristics

The population characteristics is presented in Fig 3 and S2 File. The reports of Long-acting GnRHa were submitted with an increasing trend year by year, mainly from 2019 to 2023 (Fig 3A). As for the age in reports (Fig 3B), the proportion of null was almost always the highest (except for buserelin). A higher proportion of the reports in leuprolide (23%) and goserelin (19%) were patients of 61–80 years old, while in triptorelin (15%) and histrelin (37%) was ≤ 18, and patients of 19–45 years old occupied the highest proportion (37%) in buserelin reports. In terms of role code (Fig 3C), these five

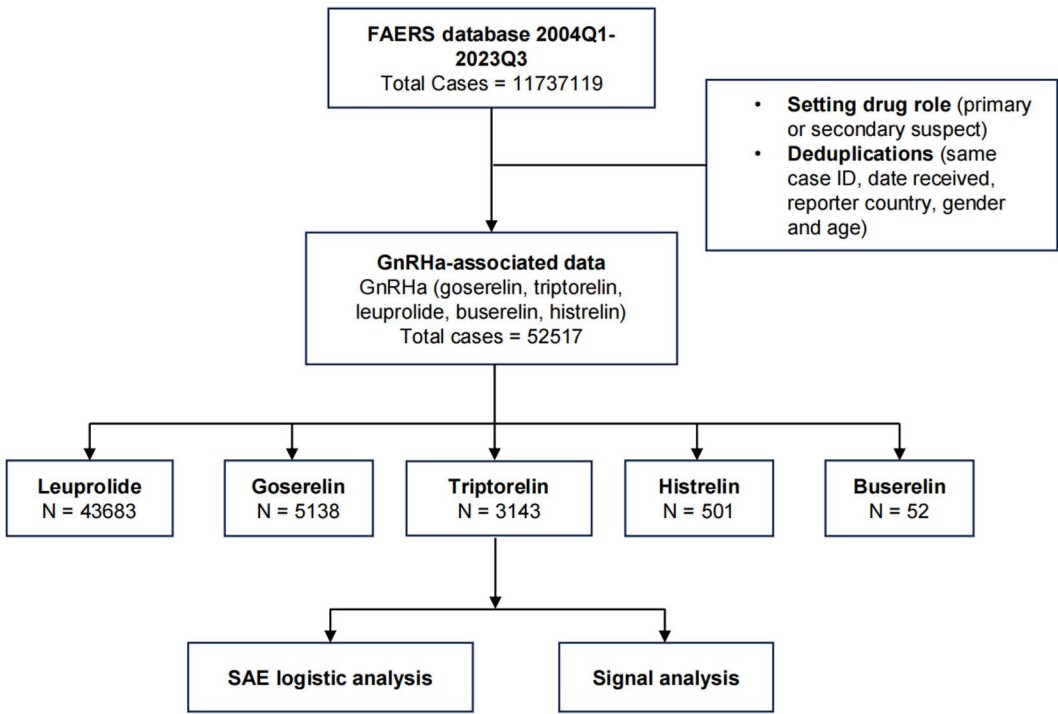

**Fig 2. The process of data processing and analysis, including the source of data acquisition, the reporting time and the cleaning.**

drugs are predominantly identified as the primary suspect drugs in the reports, with the exception of buserelin, which was consistently classified as a secondary suspect drug in 100% of the cases. As showed in Fig 1D, AEs of these five drugs were reported simultaneously in prostate cancer with leuprolide owning the highest proportion, and goserelin was as the main drug reported for breast cancer. The reports of these five drugs were mostly submitted from the United States, while reports of buserelin primarily from Germany (Fig 3E). In terms of gender (Fig 3F), the proportion of female patients in buserelin, histrelin and triptorelin was higher than that of male patients, while the opposite results occurred on goserelin, leuprolide. The observed gender disparity in GnRHa utilization patterns may primarily stems from differences in approved indications and clinical practice norms. Goserelin and leuprolide are predominantly prescribed for prostate cancer (male-prevalent), whereas buserelin, histrelin, and triptorelin are more frequently used in CPP (female-predominant) and endometriosis. This may also explain why AEs related with histrelin and triptorelin are reported in a larger proportion of patients younger than 18 years of age. Furthermore, the 3-month depot formulation of leuprolide aligns with prostate cancer management protocols requiring sustained testosterone suppression. When it comes to clinical outcomes (Fig 3G), patients receiving goserelin demonstrated significantly higher mortality (29.3%) compared to other GnRHa agents, while patients receiving buserelin demonstrated significantly higher rates of hospitalization initial or prolonged (44.2%).

## Disproportionality analysis for long-acting GnRHa

The signals unrelated to drugs or disease progression were excluded, such as general disorders and administration site conditions, product issues, social circumstances, investigations, injury, poisoning and procedural complications, congenital, familial and genetic disorders, and surgical and medical procedures. After excluding specific signals (repetitive, non-SOC included, IC025<0), the number of significant signals for each agent was as follows: leuprolide (402), goserelin (285), triptorelin (152), histrelin (35), and buserelin (19). The top twenty AEs with the highest frequency and strongest

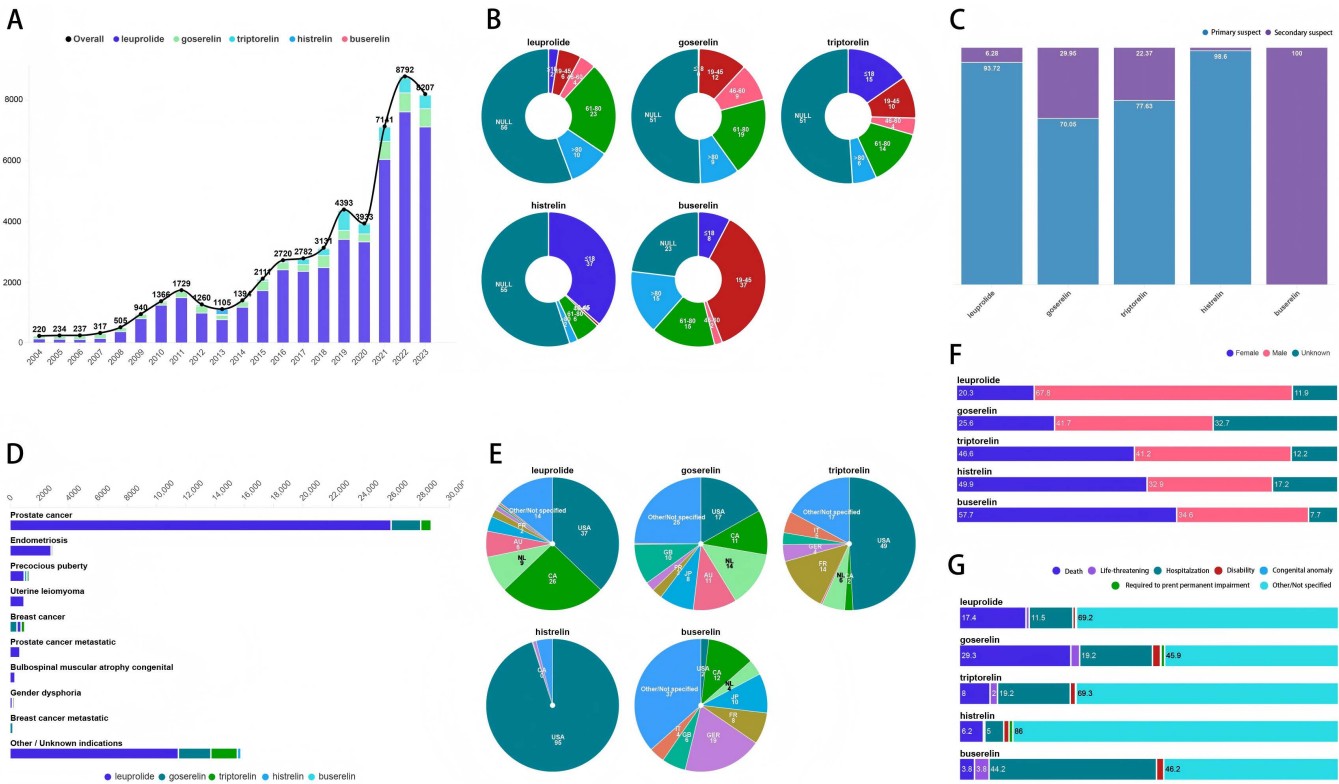

**Fig 3. Clinical characteristics of patients treated with long-acting GnRHa in the FAERS database. (A)** Date received, *p* < 0.001[a]; **(B)** Age in report, *p* < 0.001[b]; **(C)** Role code, *p* > 0.05[a]; **(D)** Top 10 indication, *p* < 0.001[a]; **(E)** Top ten reporter country, *p* < 0.001[a]; **(F)** Gender, *p* < 0.001[c]; **(G)** Outcome, *p* > 0.05[a]. a: Fisher's exact test; b: One-way ANOVA; c: Pearson Chi-Square. We have highlighted statistically significant values in boldface for emphasis. The numbers in Fig 3A and 3D represent signal number. The numbers in Fig 3B, 3C, 3E, 3F and 3G represent signal number percent.

signal intensity for these products were analyzed. As shown in Fig 4 and S3 File, the most frequently reported AE with leuprolide was hot flush, whereas the most common AE with goserelin was malignant neoplasm progression. Ovarian hyperstimulation syndrome (OHSS) was the most frequently reported AE for both triptorelin and buserelin, whereas cutaneous rash emerged as the predominant AE associated with histrelin. The AEs with the strongest signal intensities were as follows: urinary tract toxicity (ROR = 490.89) for leuprolide, malignant spinal cord compression (ROR = 3958.65) for goserelin, ovarian hyperstimulation syndrome (ROR = 719.85) for triptorelin, delayed menarche (ROR = 16901.8) for histrelin, and noninfective oophoritis (ROR = 26614.6) for buserelin. Likewise, as shown in Fig 4 and S3 File, ovarian hyperstimulation syndrome (n = 181, ROR = 719.85) demonstrated both a prevalence and a robust signal strength for triptorelin and buserelin.

To further analyze the differences among the five long-acting GnRHa agents, the comprehensive signal mining results at the SOC level are shown in Fig 5 and S4 File. As shown in Fig 5, the proportion of AEs in each SOC system was different. Reproductive system and breast disorders possessed the greatest proportion in AEs of the five drugs, followed by neoplasms benign, malignant and unspecified (including cysts and polyps), renal and urinary disorders, musculoskeletal and connective tissue disorders and nervous system disorders. A greater proportion of AEs in the neoplasms benign, malignant, and unspecified (including cysts and polyps) was also exhibited by leuprolide, goserelin, and triptorelin. In addition to the reproductive system and breast disorders mentioned above, buserelin also displayed a higher percentage

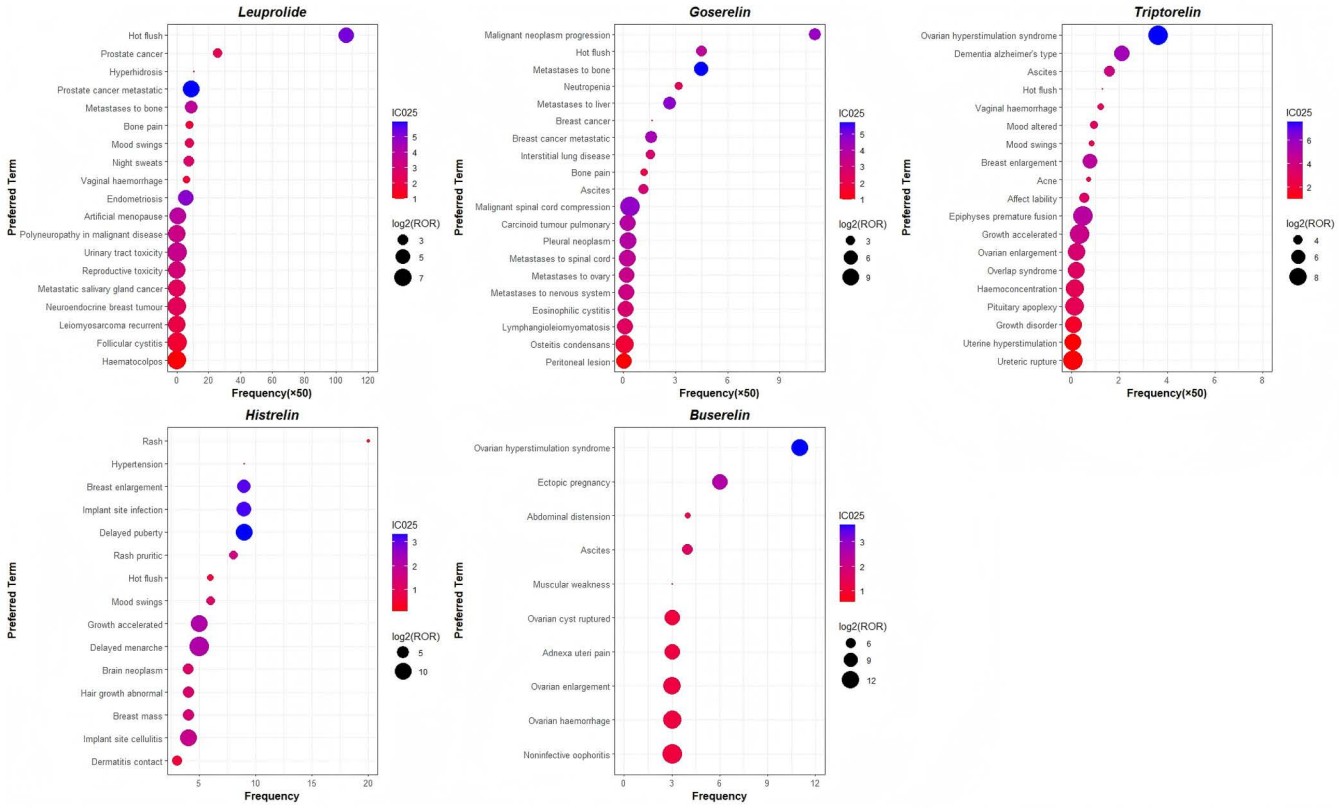

**Fig 4. Union of the top 20 signals with the highest frequency or ROR value for long-acting GnRHa, including leuprolide, goserelin, triptorelin, histrelin, buserelin.** The x-axis displays the normalized AE frequency (original values divided by 50) for leuprolide, goserelin, and triptorelin, while the y-axis represents the different AEs. The bubble size reflects the ROR value with log2 (ROR). We have presented the top twenty AEs with the highest frequency using bubbles, where each bubble corresponds to an AE, appearing on the far right of the x-axis, and the top twenty AEs with the highest signal strength are shown in the biggest bubbles. And AEs that display both high frequency and strong intensity are shown in big blue bubbles appearing to the right side of the x-axis.

of AEs in the gastrointestinal disorders. While histrelin displayed a higher percentage of AEs in psychiatric disorders and skin and subcutaneous tissue disorders, which was the most different case compared to the other four products.

## Serious adverse events/Important medical events linked to long-acting GnRHa

Serious adverse events/Important medical events were significant factors that deserve our special attention because of restrictions on the widespread use of drugs. An Important Medical Event (IME) is a clinical event that does not meet the stringent criteria for a serious adverse event (SAE) but has the potential to be clinically significant and requires proactive medical intervention to avoid a deterioration or serious outcome (such as hospitalization, disability, or death). IME is also a document in the form of a list of updated versions regularly published on EMA's official website (https://www.ema.europa.eu/en/human-regulatory/research-development/pharmacovigilance/eudravigilance/eudravigilance-system-overview). Univariate and multivariate logistic regression analyses were conducted to determine the odds ratio (OR) of IMEs associated with long-acting GnRHa across various exposure factors. The results presented in Table 1 indicated that "drug, gender, off-label use and age" were independent factors that significantly influence IMEs associated with long-acting GnRHa. As shown in Table 1, compared with patients receiving leuprolide, the risk of IMEs was 27.94 times higher in patients with buserelin, 6.78 times higher with goserelin, 1.22 times higher in histrelin, and 3.58 times higher in triptorelin. Furthermore,

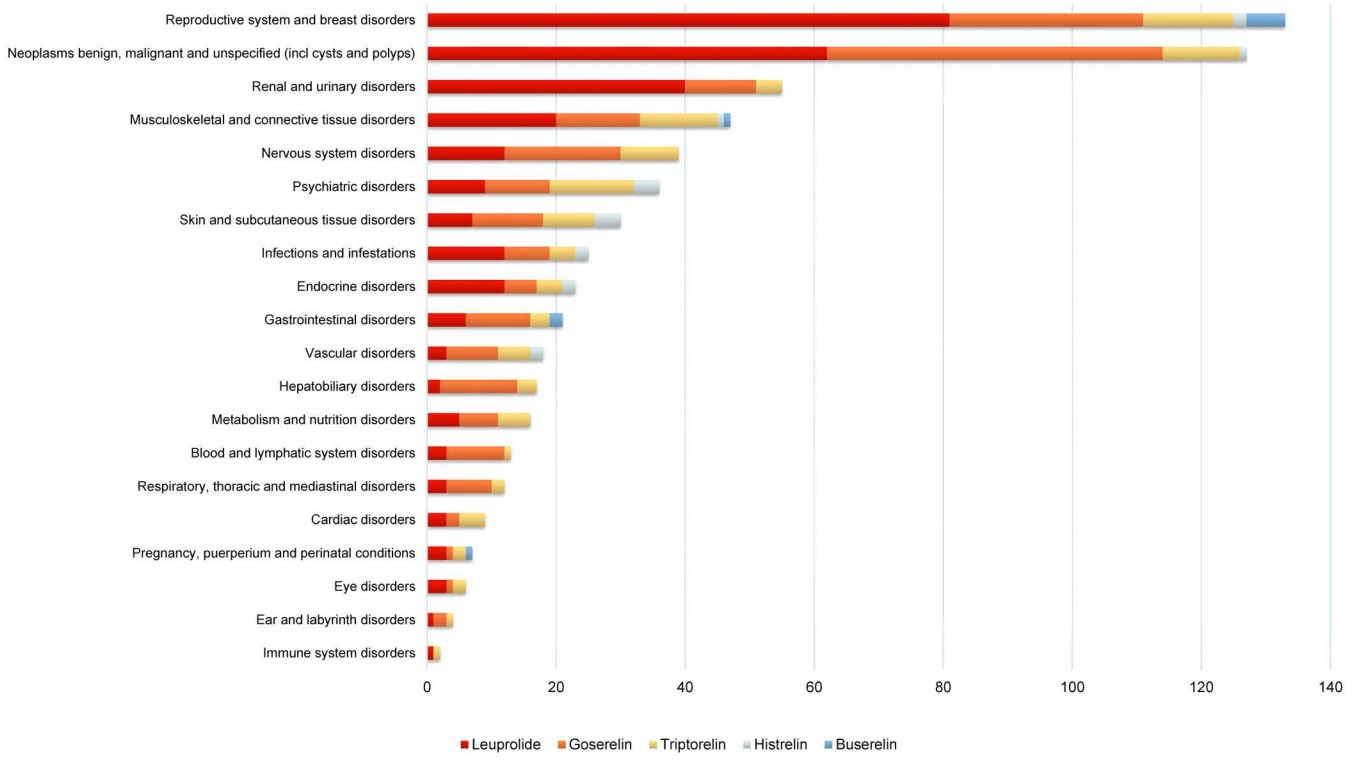

**Fig 5. The signal number of long-acting GnRHa-related significant Adverse Events (AEs) at the System Organ Class (SOC) level.**

females and males exhibited comparable risks of IMEs. Patients using long-acting GnRHa off-label use had a higher risk of IMEs with 1.40 times compared to those on-label use. Compared to the reference group (aged 19–45 years), the risk of IMEs with long-acting GnRHa was significantly lower in patients under 18 years (risk ratio: 0.29), while no significant differences were observed in older age groups (46–60, 61–80, and >80 years).

## Discussion

### Reproductive system and breast disorders

Reproductive system and breast disorders exhibited the highest incidence among AEs across the five long-acting GnRHa agents, though with notable inter-drug variations, become the most concerned SOC due to the hormonal effects of long-acting GnRHa [25,26]. Our research showed that amenorrhoea (n = 23, ROR = 5.73) was the most common AE for goserelin, ovarian hyperstimulation syndrome for triptorelin (n = 181, ROR = 719.851) and buserelin (n = 11, ROR = 2700.39), vaginal haemorrhage (n = 305, ROR = 3.48) for leuprolide, breast enlargement (n = 9, ROR = 103.55) for histrelin, in reproductive and breast systems.

However, it is interesting to note that GnRHa had been reported to have some favorable effects on ovarian. The study conducted by Senra J C et al. (2018) revealed that women who received GnRHa in addition to chemotherapy exhibited a higher rate of spontaneous pregnancy after treatment completion, when compared to those who solely underwent chemotherapy [27]. Furthermore, gonadotropin-releasing hormone agonists induce estrogen suppression in premenopausal women through temporary ovarian suppression, which reduces the growth of breast tumors and disease recurrence. Currently, GnRHa therapy is primarily recommended for the preservation of ovarian function and fertility in premenopausal patients, as well as for reducing the occurrence of adverse events [2,28].

**Table 1. Univariate and multivariate logistic regression analyses of the odds ratio for long-acting GnRHa-related important medical events.**

| Variable | Factor | Univariate analysis | | Multivariate analysis | |
|---|---|---|---|---|---|
| | | OR (95%CI) | P | OR (95%CI) | P |
| Drug | leuprolide | Ref | / | Ref | / |
| | buserelin | 24.77 (5.97,102.68) | <0.001 | 27.94 (6.62,118.02) | <0.001 |
| | goserelin | 6.93 (6.21,7.74) | <0.001 | 6.78 (6.06,7.59) | <0.001 |
| | histrelin | 0.45 (0.33,0.61) | <0.001 | 1.22 (0.87,1.70) | 0.254 |
| | triptorelin | 2.39 (2.14,2.67) | <0.001 | 3.58 (3.16,4.06) | <0.001 |
| Gender | Female | Ref | / | Ref | / |
| | Male | 1.19 (1.12,1.26) | <0.001 | 1.11 (0.97,1.27) | 0.138 |
| Off-label use | No | Ref | / | Ref | / |
| | Yes | 1.35 (1.27,1.43) | <0.001 | 1.40 (1.31,1.50) | <0.001 |
| Age group | 19-45 | Ref | / | Ref | / |
| | <=18 | 0.32 (0.28,0.36) | <0.001 | 0.29 (0.25,0.33) | <0.001 |
| | >80 | 1.41 (1.29,1.54) | <0.001 | 1.83 (1.56,2.15) | <0.001 |
| | 46-60 | 1.14 (1.02,1.27) | 0.017 | 1.16 (1.01,1.33) | 0.030 |
| | 61-80 | 0.88 (0.81,0.95) | 0.001 | 1.11 (0.95,1.30) | 0.180 |

Ref: Reference; OR: Odds Ratio; CI: Confidential Interval. Statistically significant values are marked in boldface.

It can be seen that the pharmacological effects of long-acting GnRHa on the ovary coexist with adverse reactions. Therefore, when using such drugs, it is necessary to strictly monitor whether the indicators of reproductive and breast system are normal.

## Neoplasms benign, malignant and unspecified (including cysts and polyps)

AEs in neoplasms benign, malignant and unspecified (including cysts and polyps), following administration of leuprolide, goserelin, and triptorelin, have been thought to be strongly associated with prostate cancer, since prostate cancer is the predominant indication for leuprolide, goserelin, and triptorelin, comprising 59.60%, 39.37%, and 22.88% of their respective utilization profiles. However, there are some special cases that should not be ignored. We reviewed the goserelin drug insert on the FDA website and found that "Tumor Flare Phenomenon" is one of the "WARNINGS AND PRECAUTIONS", which draws the attention: transient worsening of tumor symptoms may occur during the first few weeks of treatment with goserelin, which may include ureteral obstruction and spinal cord compression.

In view of this, it is very important to monitor patients at risk for complications of tumor flare after receiving leuprolide, goserelin and triptorelin.

## Renal and urinary disorders

Our research showed that dysuria was the most common AE for both goserelin and leuprolide, haematuria for triptorelin in renal and urinary systems. Urinary tract toxicity was the strongest AE in leuprolide, and no adverse events associated with this system were reported in buserelin.

## Musculoskeletal and connective tissue disorders

AEs within the musculoskeletal and connective tissue systems have been documented in multiple clinical studies, one of the side effects of GnRHa was a decrease in bone mineral density [29–31]. Our research showed that bone pain was the most common AE for both goserelin and leuprolide, premature fusion of epiphysis for triptorelin, muscular weakness for buserelin, and growth accelerated for histrelin in musculoskeletal and connective tissue disorders.

## Nervous system disorders

The most common AE for goserelin, triptorelin and leuprolide respectively was peripheral neuropathy (n = 50, ROR = 2.55), Alzheimer's type dementia (n = 106, ROR = 81.06), embolic stroke (n = 29, ROR = 3.95). Nevertheless, no adverse events associated with this system were reported in buserelin and histrelin in our statistics.

## Psychiatric disorders

According to the drug instructions for long-acting GnRHa, mild to moderate adverse reactions are frequently reported, whereas severe events are rare. In our cohort analysis of five drugs, mood-related adverse events were predominantly associated with four agents (leuprolide, goserelin, triptorelin, and nafarelin), manifesting as mood swings and mood altered events. In contrast, buserelin showed no significant association with psychiatric adverse effects.

Frokjaer VG et al. (2020) suggested that the transition of sex hormones, especially sharp decrease in estradiol level, may lead to severe depressive episodes in certain women [32]. Their research also indicated that the GnRha model demonstrated an estradiol-dependent depressive response in healthy women, and heightened sensitivity to estrogen signaling at the gene expression level may contribute to an increased risk for depressive symptoms during short-term manipulation of sex hormones with GnRHa.

However, a study by Wagner-Schuman M et al. (2023) demonstrated the benefits of GnRHa in mood effects. Their study shown that the use of GnRHa with hormone addback should be considered as a treatment for premenstrual dysphoric disorder (PMDD) when first- and second-line treatments, such as selective serotonin reuptake inhibitors (SSRIs) and oral contraceptives, have failed to provide symptom relief [33]. As a result of these comprehensive studies, there is a need for improved education on emotional cognition and mental functioning.

The situation of long-acting GnRHa in psychiatric system is the same as that in reproductive and breast system: the benefits coexist with the adverse effects.

## Skin and subcutaneous tissue disorders

In previous studies, long-acting GnRHa had often been reported to skin-related adverse reactions, such as estrogen deprivation symptoms and oily skin [34], injection site granulomas [35], necrotic skin ulceration [36], disseminated urticarial rash [12], erythema nodosum [37] and drug-induced vasculitis [38].

Besides local reactions, systemic hypersensitivity reactions such as urticaria, anaphylaxis, serum sickness and Henoch-Schönlein purpura (HSP) have been reported during GnRHa treatment in Central Precocious Puberty [39]. Most of these adverse events are associated with intramuscular injections.

Prior findings from our research show that histrelin displayed a higher percentage of AEs in skin and subcutaneous tissue disorders with "rash" being the highest frequency AE. Other medications differ in that the most prevalent AE for goserelin was "night sweats", "hair growth abnormal" for triptorelin, and "hyperhidrosis" for leuprolide. Notably, no adverse events related to skin and subcutaneous tissue disorders were documented in reports of buserelin acetate.

If patients suffer from subcutaneous nodules, urologists should consider changing the drug to another type of luteinizing hormone-releasing hormone analogues such as goserelin acetate.

### Gastrointestinal disorders

During treatment with GnRH analogs, a number of patients experience gastrointestinal side effects, with a small percentage developing severe complications such as chronic intestinal pseudo-obstruction (CIPO) or enteric dysmotility (ED). These conditions are characterized by degenerative and inflammatory changes in the GI tract [40], though their pathogenesis remains incompletely understood. Notably, Ohlsson B et al. (2007) reported a unique case of a 30-year-old woman who developed CIPO following buserelin administration, where the mechanism was specifically attributed to anti-GnRH antibody formation leading to destruction of GnRH-producing neurons in the myenteric plexus [41]. While this case highlights a potential immunogenic pathway for severe complications, it should be noted that such antibody-mediated effects have not been widely documented across the GnRH analog class.

Contrasting with these rare adverse events, emerging preclinical and clinical evidence suggests that GnRHa may paradoxically benefit women with menstrual cycle-related irritable bowel syndrome (IBS). Clinical trials have demonstrated therapeutic potential for alleviating characteristic symptoms including nausea, vomiting, bloating, abdominal pain, and early satiety, as well as improving overall gastrointestinal symptom burden [42,43].

### Vascular disorders

As mentioned earlier, our research found that the most prevalent AE for leuprolide, goserelin, histrelin and triptorelin was hot flush. Histrelin reported less on vascular disorders, just hypertension and hot flush. Goserelin had the most reported cardiovascular adverse events: hot flush, arterial haemorrhage, venous limb thrombosis, lymphoedema, arterial thrombosis, haemorrhagic shock, blood pressure fluctuation.

There were no vascular disorders reported in buserelin's adverse events after 2004, as adverse events reported in the FAERS data were from 2004 and later. In 1991, Pinto S et al. found that an increased risk of cardiovascular disease had been found in postmenopausal women in comparison to premenopausal women and buserelin treatment induced a condition of increased thrombotic risk [44].

In addition, some studies identified an increased risk of incident coronary heart disease, myocardial infarction, and cardiovascular death among men with prostate cancer treated with a GnRHa [45,46]. GnRHa therapy in central precocious puberty (CPP) may cause metabolic disturbances that generate increased risk of diabetes mellitus (DM) and cardiovascular disease (CVD) [47].

Regarding why the use of GnRHa is more likely to cause vascular disorders, more than twenty years ago, Dockery F et al. (2003) once gave an explanation that loss of androgens in men lead to an increase in aortic stiffness and serum insulin levels, and may therefore adversely affect cardiovascular risk [2,48].

To sum up, pharmacovigilance data suggested that GnRHa are associated with cardiovascular risks. We propose that pharmacists should provide counseling to these patients on primary disease prevention. Men receiving androgen deprivation therapy (ADT) should be monitored routinely for signs and symptoms of metabolic syndrome, diabetes, and coronary artery disease (CAD). Healthy lifestyle practices should be encouraged, and physical therapy should be considered for these patients [49]. Clinicians should consider these data when prescribing hypertension especially in patients with cardiovascular comorbidities [46].

### Serious adverse events/Important medical events

Overall, the adverse events reported on GnRHa are generally mild. Patients under the age of 18 carries the lowest risk, which bore out the reason why long-acting GnRHa is the gold standard in the treatment of CPP [50]. As for patients with buserelin, they took possession of the greatest risk of IMEs, which is worth further study. The higher IME rate with buserelin, despite its favorable overall AE profile, underscores the importance of severity-based drug safety assessments. Clinicians should consider both AE frequency and potential for rare but catastrophic events, particularly in high-risk populations (e.g., patients with autoimmune histories).

## Limitations

The following notable limitations are inherent in our study: (1) The reports are predominantly sourced from America and Europe, with a scarcity of data from Asian or African regions. (2) The ROR solely signifies an elevated risk of reported AEs and does not mirror the actual clinical risk. (3) The spontaneous reports within the FAERS database may suffer from quantitative biases and incomplete documentation. (4) The lack of longitudinal exposure data limits the ability to estimate age-adjusted morbidity and mortality rates, thereby precluding subgroup analyses and causal inference. (5)Diseases themselves can also bring adverse events. (6) There may be differences in the timing of the launch of the five drugs, so there may also be differences in the reporting of adverse events as a result.

## Conclusions

In this study, we found that the reported adverse events were basically consistent with the drug label. In our study, up to September 2023, the following number of reports were submitted to FAERS for each drug: 43,683 reports for leuprolide, 5,138 for goserelin, 3,143 for triptorelin, 501 for histrelin, and 52 for buserelin. Ultimately, a comprehensive analysis and comparative assessment were conducted, revealing 402 significant signals for leuprolide, 285 for goserelin, 152 for triptorelin, 35 for histrelin, and 19 for buserelin. Furthermore, the effects of various exposure parameters were examined by us on important medical events and contrasted the proportion of significant AEs among the five long-acting GnRHa across the nine systems. However, our research indicated that there were adverse event profile differences in the major systems of the five drugs. What makes us think deeply is that the therapeutic effects of long-acting GnRHa in psychiatric system and reproductive and breast system coexist with the adverse effects. The results of our analysis better illustrate the importance of actual pharmacovigilance studies. In summary, significant disparities exist between the adverse event profiles of long-acting GnRHa, affected by drug routes and pharmacological actions. As a result, drug safety management should consider the disease, the drug itself (e.g., route of administration), age and other factors, so as to monitor the drug according to these characteristics.

Collaborative efforts among healthcare providers, society, and patients through enhanced surveillance, education, regulations, and feedback systems are expected to mitigate adverse drug reactions. In the future, we will focus on the relationship between route of administration, dose and AEs related with long-acting GnRHa.

## Supporting information

**S1 Table. The corresponding algorithms, equations and criteria**
(DOCX)

**S2 File. The population characteristics**
(XLSX)

**S3 File. The top twenty AEs with the highest frequency and strongest signal intensity**
(XLSX)

**S4 File. The comprehensive signal mining results at the SOC level**
(XLSX)

## Author contributions

**Conceptualization:** Yuting Chen, Weitao Lu.

**Data curation:** Yuting Chen, Weitao Lu, Ximin Zhang.

**Formal analysis:** Yuting Chen.

**Funding acquisition:** Yuting Chen, Huancun Feng.

**Investigation:** Yuting Chen, Ruilian Liao, Ximin Zhang, Wang Chen, Jing Wang.

**Methodology:** Yuting Chen, Wang Chen, Jing Wang.

**Project administration:** Yuting Chen, Ruilian Liao.

**Resources:** Yuting Chen, Weitao Lu, Huancun Feng.

**Software:** Yuting Chen, Ximin Zhang.

**Supervision:** Yuting Chen.

**Validation:** Yuting Chen, Ruilian Liao.

**Visualization:** Yuting Chen, Ruilian Liao.

**Writing – original draft:** Yuting Chen, Weitao Lu.

**Writing – review & editing:** Weitao Lu, Huancun Feng.

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
