## [Decision Letter · Decision Letter 0]

PONE-D-24-54304Adverse Event Profile Differences among Long-acting Gonadotropin-releasing hormone analogs: A Real-world, Pharmacovigilance StudyPLOS ONE

Dear Dr. Feng,

Thank you for submitting your manuscript to PLOS ONE. After careful consideration, we feel that it has merit but does not fully meet PLOS ONE’s publication criteria as it currently stands. Therefore, we invite you to submit a revised version of the manuscript that addresses the points raised during the review process.

Please submit your revised manuscript by Mar 27 2025 11:59PM. If you will need more time than this to complete your revisions, please reply to this message or contact the journal office at plosone@plos.org . Please include the following items when submitting your revised manuscript:

We look forward to receiving your revised manuscript.

Kind regards,

Arturo Aguilar-Rojas

Academic Editor

PLOS ONE

**Journal Requirements:**

This work was supported by 2024 PSM Guangdong Pharmaceutical Science Research Fund No.2024KP106) and 2024 Guangdong Provincial universities characteristic innovation project (No.2024KTSCX078).

Reviewers' comments:

Reviewer's Responses to Questions

**Comments to the Author**

1. Is the manuscript technically sound, and do the data support the conclusions?

Reviewer #1: Yes

Reviewer #2: Partly

2. Has the statistical analysis been performed appropriately and rigorously? 

Reviewer #1: Yes

Reviewer #2: Yes

3. Have the authors made all data underlying the findings in their manuscript fully available?

Reviewer #1: Yes

Reviewer #2: Yes

4. Is the manuscript presented in an intelligible fashion and written in standard English?

Reviewer #1: Yes

Reviewer #2: No

5. Review Comments to the Author

**Reviewer #1: ** Chen et al entitle “Adverse Event Profile Differences among Long-acting Gonadotropin-

releasing hormone analogs: A Real-world, Pharmacovigilance Study” clearly reported that long-acting GnRHa could be adverse drug for prostate cancer, breast cancer, endometriosis, uterine leiomyomas, and central precocious puberty.

Why are you using the FAERS database to study this drug, it should be that someone else has done this with this similar database similar methodology, so you can write cite in the INTRODUCTION section about some specific other similar studies, such as recommending a few (It is equivalent to saying that someone else has done this type of research using the FAERS database, and you can use this database to do research related to long-acting GnRHa as well):【1】Wang Y, Zhao B, Yang H, Wan Z. A real-world pharmacovigilance study of FDA adverse event reporting system events for sildenafil. Andrology. 2024 May;12(4):785-792. doi: 10.1111/andr.13533. Epub 2023 Sep 19. PMID: 37724699.【2】Zhao B, Fu Y, Cui S, Chen X, Liu S, Luo L. A real-world disproportionality analysis of Everolimus: data mining of the public version of FDA adverse event reporting system. Front Pharmacol. 2024 Mar 12;15:1333662. doi: 10.3389/fphar.2024.1333662. PMID: 38533254; PMCID: PMC10964017.【3】Yang H, Wan Z, Chen M, Zhang X, Cui W, Zhao B. A real-world data analysis of topotecan in the FDA Adverse Event Reporting System (FAERS) database. Expert Opin Drug Metab Toxicol. 2023 Apr;19(4):217-223. doi: 10.1080/17425255.2023.2219390. Epub 2023 May 30. PMID: 37243615.【4】Zhao B, Zhang X, Chen M, Wang Y. A real-world data analysis of acetylsalicylic acid in FDA Adverse Event Reporting System (FAERS) database. Expert Opin Drug Metab Toxicol. 2023 Jan-Jun;19(6):381-387. doi: 10.1080/17425255.2023.2235267. Epub 2023 Jul 12. PMID: 37421631.【5】Li, Jie, Zhao, Bin, Zhu, YongQing, Wu, Jibiao, Vitreoretinal Traction Syndrome, Nitrituria and Human Epidermal Growth Factor Receptor Negative Might Occur in the Aromatase-Inhibitor Anastrozole Treatment, International Journal of Clinical Practice, 2024, 5132916, 9 pages, 2024. https://doi.org/10.1155/2024/5132916【6】Zhong, C., Zheng, Q., Zhao, B., & Ren, T. (2024). A real-world pharmacovigilance study using disproportionality analysis of United States Food and Drug Administration Adverse Event Reporting System events for vinca alkaloids: comparing vinorelbine and Vincristine. Expert Opinion on Drug Safety, 23(11), 1427–1437. https://doi.org/10.1080/14740338.2024.2410436

Besides,

I have general suggestion is:

1) Can Author put a bar chart for the percentage of long-acting GnRHa uses in five country and hospitalization, death, life threating events, and disability.

2) I also encourage to author please put the bar chart for table where author indicated that prostate cancer, breast cancer, endometriosis, uterine leiomyomas, and central precocious puberty related adverse event for use of long-acting GnRHa.

3) I also encourage the author please use 1 graphical representation of long-acting GnRHa and their adverse event (long-acting GnRHa affecting which organ in human body)

**Reviewer #2: ** The aim of the study was to investigate the adverse event (AE) profile differences of long-acting GnRHa reported by the US Food and Drug Administration Adverse Event Reporting System (FAERS).

General comments

The context for the study was unclear, there needs to be background that included definition of and the function of the GNRH analogues and long acting GNRH analogues to give context for the discussion. The study results and discussion make several assumptions in this regard and needs to be very specific and clarify the scope and purpose more clearly and accurately prior to shifting focus to the adverse drug events. Occasionally it discusses issues that are more concerning short acting forms and sometimes combines these generally as analogues; there needs to a clarity clearly link discussion to the research question posed here.

The four drugs being compared needed to be systematically compared throughout the results section to demonstrate a robust report on the relevant and respective AEs. Where such data does not exist, this should be reported clearly. The results section makes reference to GnRHa related AEs but does not provide targeted information on the four drugs investigated here clearly and consistently. This reporting needs to be more organised and should include specific details where available and also what is not available to demonstrate the findings are novel and such a particular reporting is warranted. If this is addressed to be more systematically, then the paper does offer some valuable insight for researchers/health professionals.

Language and needs revision for clarity and accuracy, as there as instances of colloquial English and not appropriate for clear and accurate scientific reporting and does not meet the requirement of academic writing. There are uses of personal pronouns “we”/ “our” and contraction “What’s”, it is recommended to review this with language more appropriate for academic writing that is personal and focuses on reporting findings without emotional appeal. There are issues with no space between words/use of incorrect word form and needs to capitalise “Alzheimer's” which is a name that is derived from a specific person's name. The writing would benefit from careful proof reading or a professional editor. When authors employ author prominent style to report study data it would be appropriate to also include date of the publication. Provide a detailed description of the abbreviations throughout the manuscript.

Not sure if the authors have used a paraphraser or AI or thesaurus tools but the writing needs to reviews for clarity and accuracy, and revise word choices more appropriate for the context of communication here. The language and grammar errors significantly detract with understanding the information being communicated in this paper.

Specific comments

Abstract

“Reproductive system and breast disorders possessed the greatest proportion in AEs of the five long-acting GnRHa.” – review word choice “possessed”

“Buserelin and histrelin demonstrated a higher

percentage of AEs in the gastrointestinal and psychiatric disorders separately, which also exhibited a higher risk of important medical events in logistic regression analysis and the risk was lower in patients under 18 years old than those over 18 .” – Vary sentence structure and revise for clarity and accuracy.

Introduction

Paragraph 1

“exacerbates, a range of hormone-dependent diseases” - vague, identify the diseases, as the risk is not equal/comparable

“LHRH”- provide a detailed description of the abbreviation at first use

“puberty, endometriosis, adenomyosis etc.,” – complete the list appropriately, not academic convention

“also for the prevention of chemotherapy-induced premature ovarian failure in cancer women[2-5].” – revise expression “cancer woman”

Paragraph 2: Identify adverse reactions with appropriate type of GnRHa. Are these findings relevant to particular use/dosage of GnRHa administered?

“CPP” - provide a detailed description of the abbreviation at first use

“hypertension (HTN)[10]” - is this an accepted abbreviation?

“had been covered during the use of GnRHa” - not clear what this claim means, revise language

“As one of the effective drugs for treating prostate cancer, the adverse reactions reported including life-threatening anaphylaxis[11], follicular mucinosis and mycosis-fungoides-like drug” - Not clear which drug/s the authors are implying here, provide details.

“…follicular mucinosis and mycosis-fungoides-like drug eruption[12], cardiovascular (CV)[13].”- revise for grammar and clarity, what aspect of cardiovascular does the authors imply here?

Paragraph 3

“Therefore, we utilize FAERS to evaluate the safety characteristics of long-acting GnRHa and provide a practical and secure reference for their clinical applications.”- review for grammar, utilised/utilized (past tense)

Materials and Methods

Data sources

“and therapeutic biologic products” – review grammar, “biological”

Statistical analysis

Paragraph 1

“…is one of the most…”- review space between words

Paragraph 2

“…long-acting GnRHawas analyzed…” - review space between words

Results

Paragraph 1

“…Long-acting GnRHa were submitted with an increasing…”- - review space between words

“A higher proportion of the reports in leuprolide (23%) and goserelin (19%) were patients of 61-80 years old, while in triptorelin (15%) and histrelin�37%�was ≤18�and patients of 19-45 years old occupied the highest proportion�37%�in buserelin reports.” - This needs to be clarified in the introduction and provide the context for the drug administration if it is condition specific.

“In terms of role code Fig.1C�, these five drugs are essentially the primary suspect drugs in reports, except for buserelin, which was secondary suspected drugs with 100%.” - review language for clarity, claim is vague and appears to be incomplete, 100% what? Also review space between words.

“The reports of these five drugs were mostly submitted from the Americas,…” - Here are you referring to USA/ Central/South, vague word choice “Americas”.

“In terms of gender(Fig.1F), the proportion of female patients in buserelin, histrelin and triptorelin was higher than that of male patients, while the opposite results occurred on goserelin, leuprolide.” - is this context of cancer or another condition, provide specific details for results.

“goserelin has a higher proportion (29.3%) of death,…” - review language, drug cannot have death but is the cause of death.

“while buserelin has the largest percentage in hospitalization initial or prolonged (44.2%).” - review expression, fragmented sentence.

“The clinical outcomes of other drugs are distributed.” – vague claim, provide specific details for this claim, needs context.

Figure 1 legend

“…The numbers in Fig.1B, Fig.1C, Fig.1E, Fig.1F and Fig.1G represented signal number percent.” - should be "represent"

Disproportionality Analysis for long-acting GnRHa - Here authors should provide specific details as to what these specific disorders were to explain the relevance/severity of the AE being documented here. Why is this worth reporting in a publication.

Paragraph 1

“With exclusion of the signals unrelated to drugs or disease progression, such as general disorders and administration site conditions, product issues, social circumstances, investigations, injury, poisoning and procedural complications, congenital, familial and genetic disorders, and surgical and medical procedures.” - confused and incomplete sentence, please review, consider varying sentence structure here.

“Excluding the specific signals (repetitive, non-SOC included, IC025 less than 0), leuprolide, goserelin, triptorelin, histrelin and buserelin had 402, 285, 152, 35 and 19 significant signals separately.” - include data next to descriptor for clarity and accuracy of reporting.

“As shown in Fig.2 and Table S3, the most frequently reported AE for leuprolide was hot flush (n=5320), and malignant neoplasm progression(n=552) for goserelin.” - this appropriate reporting style. The authors should consider reviewing and revising their results for this style.

“…reported AE for both for triptorelin and buserelin, while rash was for histrelin.”- review expression and grammar, verbs need to be used to report observations.

“demonstrated both a prevalence and a robust signal strength for triptorelin and buserelin.” - not sure what the authors mean here. Are you saying that this syndrome was common for both the drugs as an AE?

Paragraph 2

“…triptorelin frequency with after dividing by 50),…” - review expression, confused meaning

Paragraph 3

“…long-acting GnRHa, the entire analysis results ….” – “result”, review sentence for language and grammar

“Substantially, reproductive system and breast disorders possessed the greatest proportion in AEs of the five drugs, followed by neoplasms benign, malignant and unspecified (incl cysts and polyps),…” – delete “substantially”, incorrect use, review sentence for grammar; “incl” use complete words for academic writing.

SAE/Important medical events linked to long-acting GnRHa

“SAE/Important medical events (IMEs) were significant factors that…” - These IME needs to be defined in the introduction to explain the relevance of the results and give clear context for the analysis.

“drugs.Univariate” – review spaces between words

“Additionally, females exhibited a similar risk of IMEs in males.” - confused meaning, review for grammar and language.

“…off-label use…” - what are these, define these terms in the introduction, to given context and links for the relevance of these results.

“…over 80 had a approximate risk.” - not sure what the authors mean here. review for grammar and expression, clarify with specific details as to what the authors mean “approximate”.

Discussion

Reproductive system and breast disorders

Paragraph 1 : are these a general observation, are there different observations for each of the four drugs being investigated here. Needs to clarify or provide the scope of the findings here.

“…and buserelin(n=11, ROR=2700.39), vaginal haemorrhage(n=305, ROR=3.48) for leuprolide, breast enlargemen(n=9, ROR=103.55)) for histrelin, in reproductive and breast systems.” - Confused expression, review as done for previous two drugs. Review spaces between words, review parenthesis here.

“conducted by Senra J C et al.” – include date to publication

“…completion, as compared…” – replace as with appropriate transition (when)

“What’s more, in premenopausal women, GnRHa suppress estrogen …” – contractions are not appropriate for academic writing.

Paragraph 2

“Therefore, when using such drugs, it is necessary to strictly monitor whether the indicators of reproductive and breast system are normal.” - Is there data to report whether there are dose specific differences in these observations?

Neoplasms benign, malignant and unspecified (incl cysts and polyps): Are these reversible changes? Here does dosage and frequency impact the outcomes. Does the author have data/statistical evidence to report this claim.

Paragraph 1

“…unspecified (incl cysts and polyps) induced by leuprolide, goserelin and triptorelin, we consider that it is highly correlated…” – “incl” is unclear abbreviation, “we” pronouns are not appropriate for academic reporting, “is”, delete; review for grammar and language

“…in the three drugs with 59.60%,39.37% and 22.88% severally.” – incorrect use of “severally”

Musculoskeletal and connective tissue disorders

“In musculoskeletal and connective system of quite a few reports.” – vague, review language and expression.

“Our research showed that bone pain was the most common AE for both goserelin and leuprolide, epiphyses premature fusion for triptorelin, muscular weakness for buserelin, and growth accelerated for histrelin in musculoskeletal and connective tissue disorders.” - premature fusion of epiphysis (review spelling and expression).

Nervous system disorders

“…respectively was neuropathy peripheral (n=50, ROR=2.55), dementia alzheimer's type (n=106, ROR=81.06), embolic stroke (n=29, ROR=3.95). Nevertheless, no adverse events associated with this system were reported in buserelin and histrelin in our statistics.” – “neuropathy peripheral”, peripheral neuropathy, adjective needs to come before the noun; “dementia alzheimer's type” see previous comment, adjective needs to come before the noun; “our statistics” Should be results.

Psychiatric disorders

Paragraph 1

“In the instructions for this type of drugs, such adverse reactions are more common but less serious.” – vague expression, not cleat what the author referring to here, review to be more specific here.

“In our study, except for buserelin, the other four drugs involved only mood swings, mood altered events.” – review for language and clarity.

Paragraph 2

“Frokjaer VG et al. suggested” – include date for publication

“…that the transition of sex hormones…” - not clear what the author is referring to here. In what way are the hormones transitioning.

“… during short-term manipulation…” – would this be increase/decrease in the levels of hormones/ dose dependent effects?

Paragraph 3

“However, we were surprised to find …” - avoid emotive language in reporting to prevent bias and to be objective in reporting.

“… GnRHa with hormone addback…” - not clear what this phrase means here

“SSRIs” - provide description for abbreviation

“ Wagner-Schuman M et al. demonstrating its favorable mood effects[28].” – include date for publication, What does the authors mean by favourable, provide specific details to support the claims.

Paragraph 4

“psychiatric” – term is in bold, unclear why

Skin and subcutaneous tissue disorders

Paragraph 1

“…skin-related adverse reactions, such as estrogen deprivation symptoms and oily skin…” – is this a reference to endogenous and or exogenous estrogen levels, provide specific details as to how this supports the findings of this paper.

“…erythema nodosum[32] , drug-induced vasculitis…” – include “and at the end of list

Paragraph 2

“The previous of our results show that histrelin…” - review expression for grammar and language

“By the same coincidence, no adverse events associated with this system …” - review for language and expression

“If patients suffer from subcutaneous nodules, urologists should consider changing the drug to another types of luteinizing hormone-releasing hormone analogues such as goserelin acetate.” - should be type (singular), is there literature to support this finding/general practice/recommendation?

Gastrointestinal disorders

“Ohlsson B et al. reported a case of a 30-year-old woman who developed CIPO as a result of buserelin-induced formation of anti-GnRH antibodies leading to destruction of GnRH-producing neurons in the myenteric plexus.[36]. - There needs to be this type of reporting to given context for these observations as much of this cannot be blanket AEs.

“However, the case of their beneficial effects on the gastrointestinal tract deserves our attention. Gondotropin-releasing hormone agonist administration is effective in…” - avoid such superficial and emotive claims as it detracts from the reported results.

- is there a particular drug type that is recommended and what is the recommended or therapeutic dosage in the treatment of this group of disorders. This information would add value to the manuscript and also some context for these reporting.

Vascular disorders: vague reporting, provide some details to specify the severity of these AEs

Paragraph 1

“flush, in fact, also applies to goserelin, histrelin and triptorelin of vascular disorders.” - review for grammar and expression.

“venous thrombosis limb,” “ shock haemorragic”– review expressions

Paragraph 2

“disturbances that confer increased risk of (CVD) diabetes mellitus (DM) and cardiovascular disease” –“ confer”, review word choice; “increased risk of (CVD)” , provide description for abbreviation and incorrect use of parenthesis; “and cardiovascular disease”, confused use here.

Paragraph 3

“As for the principle” – confused, not sure what the authors mean here.

“Dockery F et al. once gave an explanation that loss of androgens…” – include date for publication, is this finding relevant to current practice?

“…adversely affect cardiovascular risk twenty years ago[2, 43].” – revise for language and expression

Paragraph 4

“Therefore, the cardiovascular harm of GnRHa is also not to be underestimated” - revise for language and expression

- Recommendation needs to be supported with evidence from clinical literature as best practice/current practice.

“prescribing HT especially” - provide a detailed description of the abbreviation at first use.

SAE/Important medical events

“buserelin, they took” – review space between words

Limitations

“2) The ROR solely signifies an elevated risk of reported AEs and does not mirror the actual clinical risk.” - This could have been discussed more explicitly in the results and discussion.

“(4) The absence of exposure data precludes the calculation of morbidity and mortality rates.” - review for language and expression

“(6) There may be differences in the timing of the launch of the five drugs, so there may also be differences in the reporting of adverse events as a result.” - a brief on the timeline for the introduction of the drugs into clinical practice is relevant to this paper could be included in the introduction.

Conclusion

“In our study, 43683, 5138, 3143, 501, and 52 reports of leuprolide, goserelin, triptorelin, histrelin and buserelin were submitted to FAERS up to September 2023. Ultimately, a comprehensive analysis and comparative assessment of 402, 285, 152, 35 and 19 significant signals for the five long-acting GnRHa were conducted.” - put data next to drug for clear reporting

“A better illustration of what an actual pharmacovigilance research entails was provided by this.”- vague claim

“actions. As a result, drug safety management should consider the disease , the drug itself (route of use), age and other factors, so as to monitor the drug according to these characteristics.” – vague claim as route of drug administration was not reported here and should have been part of the results and discussion to provide relevant context and demonstrate robust analysis here.

6. PLOS authors have the option to publish the peer review history of their article (what does this mean? ). If published, this will include your full peer review and any attached files.

**Do you want your identity to be public for this peer review?** For information about this choice, including consent withdrawal, please see our Privacy Policy .

Reviewer #1: No

Reviewer #2: No

---

## [Author Response · Author response to Decision Letter 1]

25 Mar 2025

We sincerely thank the editors and reviewers for your hard work in reviewing the manuscript. We have responded to each reviewer's comments and uploaded relevant documents as required: Revised Manuscript with Track Changes, Manuscript and Response to Reviewers. Thanks again.

---

## [Editor Report · Decision Letter 1]

PONE-D-24-54304R1Adverse Event Profile Differences among Long-acting Gonadotropin-releasing hormone analogs: A Real-world, Pharmacovigilance StudyPLOS ONE

Dear Dr. Feng, Thank you for submitting your manuscript to PLOS ONE. After careful consideration, we feel that it has merit but does not fully meet PLOS ONE’s publication criteria as it currently stands. Therefore, we invite you to submit a revised version of the manuscript that addresses the points raised during the review process. Please submit your revised manuscript by Jul 07 2025 11:59PM. If you will need more time than this to complete your revisions, please reply to this message or contact the journal office at plosone@plos.org . Please include the following items when submitting your revised manuscript:

We look forward to receiving your revised manuscript.

Kind regards,

Arturo Aguilar-Rojas

Academic Editor

PLOS ONE

Journal Requirements:

**Additional Editor Comments:**

The following are suggestions to address minor issues in the manuscript submitted by Huancun Feng.

• In the previous version, the authors did not include the figures, table, or supplementary material. Please submit a revised version that includes all necessary components to allow proper evaluation of the manuscript.

• In the following sentences, it should be clearly stated that the adverse events (AE) were observed in association with specific GnRH analogues. For example: “The AEs with the strongest signal intensities for leuprolide, goserelin, triptorelin, histrelin, and buserelin were urinary tract toxicity (ROR = 490.89), malignant spinal cord compression (ROR = 3958.65), ovarian hyperstimulation syndrome (ROR = 719.85), delayed menarche (ROR = 16901.8), and noninfective oophoritis (ROR = 26614.6), respectively.” Based on the data presented in Table S3, each of these ROR values corresponds to only one GnRHa. Please revise this paragraph to explicitly associate each AE and ROR with the specific GnRHa in which it was observed.

• In Figure 2, the authors state that the top 20 AEs are shown for each agonist. However, some graphs display fewer than 20 AE. Please clarify this discrepancy in the text or adjust the figures to consistently display 20 AE for each GnRHa.

• Define "SAE" in the title of the Results section: “SAE/Important Medical Events Linked to Long-Acting GnRHa.”

• The following sentence needs style harmonization: “Besides local reactions, systemic hypersensitivity reactions such as urticaria, anaphylaxis, serum sickness, and Henoch-Schönlein purpura (HSP) have been reported during gonadotropin-releasing hormone (GnRH) analogue treatment in Central Precocious Puberty[39]. Most of these adverse events are associated with intramuscular injections.” To maintain consistency throughout the document, remove the full hormone name and retain only the abbreviation "GnRH."

• If buserelin shows the lowest number, frequency, and intensity of AEs, please explain why this GnRHa is associated with the highest number of Important Medical Events (IME). Additionally, clearly define which criteria were used to classify events as IME or what was consider as IME in your analysis.

• In the conclusion, the authors do not indicate, based on their findings, which GnRHa might be considered the most suitable pharmacological option. In my opinion, it would be appropriate to at least suggest a preferred candidate based on the observed data.

---

## [Author Response · Author response to Decision Letter 2]

16 Jun 2025

Dear editors,

We sincerely thank you for your hard work in reviewing our manuscripts. We have revised and uploaded the relevant files one by one: manuscript, Revised Manuscript with Track Changes, Response to Reviewers. The specific modifications have also been responded to in Response to Reviewers and marked in Revised Manuscript with Track Changes.We hope that our revision work will facilitate your work. Thanks again.

---

## [Editor Report · Decision Letter 2]

Adverse Event Profile Differences among Long-acting Gonadotropin-releasing hormone analogs: A Real-world, Pharmacovigilance Study

PONE-D-24-54304R2

Dear Dr. Feng,

We’re pleased to inform you that your manuscript has been judged scientifically suitable for publication and will be formally accepted for publication once it meets all outstanding technical requirements.

Kind regards,

Arturo Aguilar-Rojas

Academic Editor

PLOS ONE

Additional Editor Comments (optional):

The authors' responses to the reviewers' comments have been satisfactory
---

## [Editor Report · Acceptance letter]

PONE-D-24-54304R2

PLOS ONE

Dear Dr. Feng,

I'm pleased to inform you that your manuscript has been deemed suitable for publication in PLOS ONE. Congratulations! Your manuscript is now being handed over to our production team.

Kind regards,

on behalf of

Dr. Arturo Aguilar-Rojas

Academic Editor

PLOS ONE